# The Pocket Park and Its Impact on the Quality of Urban Space on the Local and Supralocal Scale—Case Study of Krakow, Poland

Tomasz Bajwoluk and Piotr Langer *

Faculty of Architecture, Krakow University of Technology, Warszawska 24, 31-155 Krakow, Poland; tomasz.bajwoluk@pk.edu.pl
* Correspondence: piotr.langer@pk.edu.pl

**Abstract:** The idea of building pocket parks in cities is one of the more rational proposals for utilizing cameral spaces to create new quality in terms of green areas while accounting for the potential to blend them into the compact functiospatial structure of the contemporary city. Numerous examples of pocket park projects from around the world point to there being considerable interest in this form of greenery. The goal of this paper is to present the findings of a study of a selected number of pocket parks in Krakow, Poland, in terms of their accessibility, local determinants, and the nearby functiospatial structure, as well as whether they can be included into a wider network of service and green spaces of supralocal significance. The research method included novel field research of selected pocket parks and their surroundings. The form and function of the parks were analyzed and the type of their surrounding urban structure was determined, along with the parks' accessibility. The study investigated nine parks located in the northeastern part of Krakow in a dense development structure dominated by multi-family housing. Analyses of the parks themselves and the research on the relations and linkages between parks and their surrounding urban structure generally pointed to the accuracy of the concept of the pocket park, its universality, and its compliance with the concept of the sustainable development of urban space. The presence and manner of development of pocket parks can be said to enhance the quality of spaces in confined fragments of an urban structure and to have predominantly local significance.

**Keywords:** pocket park; functiospatial structure; urban space; greenery; accessibility

## 1. Introduction

The concentration of development in cities, transport infrastructure development, and the growth of urban centers into new areas generate an impulse to search for new green areas in cities. This is especially significant in densely developed urban structures dominated by multi-family housing. The idea of the 'pocket park' is a contemporary answer to this problem and is based on developing relatively small areas, often previously decayed, into greenery for rest and recreation purposes, along with their accompanying infrastructure and features, directly inside densely developed urban tissue [1,2]. This idea is currently a widespread and popular method of organizing housing or downtown space that is largely located in close proximity to dwellings and is thus, by design, easily accessible on the local scale.

The idea to create cameral green enclaves called pocket parks originated in North America and Europe towards the end of the 1960s. Early implementations of this idea include Paley Park in New York, US, and the Drury Lane Gardens in London, Great Britain. It stemmed from contemporaneous attempts at finding a new form of public space that would be widely accepted and oriented towards satisfying the needs of housing communities in terms of using organized, local green areas for rest and recreation. The idea

of the pocket park was also aligned with the notion of sustainable urban development by providing easy access to natural areas, including greenery, with a diverse character.

The implementation of pocket parks is aligned with the current of global and universal city development processes, including re-urbanization and the pursuit of retaining existing and stimulating the influx of new residents to urban areas and districts. New pocket parks are spaces that are desirable by local communities and play an essential role in shaping urban greenery systems. They prevent negative climate phenomena, for instance, by alleviating the urban heat island effect. The issues discussed are also closely tied with the idea of sustainable urban development, in which green areas are an integral and inseparable element of the functiospatial structure and determine the quality of the natural environment, especially in the face of growing real estate development pressure and the intensification of development in contemporary cities.

## 2. State of the Art

A review of the literature showed that research on pocket parks is multi-threaded. This arises from the widely understood significance of greenery in urban areas, as an integral element of the functiospatial structure, which considerably influences various areas of urban life. Interesting studies on pocket parks include the paper by Łabuz [3], which compared selected parks built in Krakow with similar projects in large US metropolises. Studies in a similar vein are conducted by Iorpenda et al. in relation to pocket parks in the Jos metropolis in Nigeria [4], and by Luks to parks in Detroit, USA [5]. The authors point out the universality of the implementation of the idea and its positive impact on the environment, inter alia, by improving the climatic conditions in the urban environment [6–10], increasing the number of public green areas, local biodiversity [11], and sustainable water management [12]. Many researchers emphasize the functional importance of pocket parks as places for recreation and leisure [13], the promotion of health and physical activity [14–16], and education [17], and important social roles [18]. Important conclusions were reached by Hamdy and Plaku [19], as well as Abd El Aziz [20]. The authors recognize their special ecological and social role, but also their great importance in the process of revitalizing post-industrial areas [21], and even wider parts of the urban structure [22].

Many publications on the general principles of urban greenery design and siting focus on the accessibility and placement factor. This aspect was investigated by Łaszkiewicz et al. [23] in reference to Łódź and focused on access to landscaped green areas. The researchers noted that diversity in this area stems from, among others, the socioeconomic status of the residents of each district or city block. The disproportions in accessibility investigated by Łaszkiewicz et al. can become the start of a discussion on these issues, including in the context of potentially solving them at the stage of planning and revitalizing cities. The significance of green area accessibility in cities was also discussed by Xu, Haase, and Pauleit [24]. They investigated the matter in highly developed and dynamically transforming housing areas. The research area for their study was Munich and its suburban zone. They highlighted the important role of green areas in the sustainable design of large urban agglomerations, as well as in their external zones. They also noted that the development of housing is accompanied by a need to provide good access to green areas, especially notable in polycentric urban spatial structure models. Other researchers—Dei and Wang—present a parametric method that facilitates the selection of the optimal location of a pocket park in the city structure, taking into account internal connections between residential districts and the activity zones of residents [25].

A large portion of research on pocket parks focuses on the scope and design of the parks themselves and their immediate surroundings, showcasing proposals with optimal solutions that are rated the highest by their users. The findings of such a study were presented by, among others, Lee and Kim [26], Shi and Wei [27], Shahhosseini et al. [28], Mandziuk et al. [29]; they study the preferences of potential users and formulate suggestions for creating pocket parks. Note also Sinou and Kenton, who analyze the importance of environmental factors such as availability of light, acoustics, variety of greenery, and the



presence of water for the attractiveness of pocket parks [30]. Nordh and Kiersti [31] refer to the arrangement and equipment of the analyzed parks and to the functioning of their vicinity, emphasizing the usefulness of the research carried out in the design and location of new pocket parks, taking into account the need to effectively isolate these spaces from the nuisance of the environment.

Research on pocket parks in Polish cities highlights that the use of this form of urban greenery has only recently taken place in Poland. Recent studies indicate a clear relationship between the attractiveness of a pocket park and the variety of forms of its development and flexibility of use [32]. The function and form of selected pocket parks in Poland were discussed by, among others, Tokarska-Osyczka and Osyczka [33], who described suitable cases in terms of land development and the surrounding context. Some of the literature items reviewed concerned the link between the presence and accessibility of public green areas and the attractiveness of siting housing. This issue was investigated by Czembrowski [34], who analyzed selected areas of Łódź in terms of their significance and value in the city's spatial structure. Zawojska et al. [35] used the park in Wilanów to present a method of assessing the value of ecosystem services in cities and highlighted the natural and economic potential of urban public greenery areas and their multidimensional worth. Another interesting research trajectory associated with the subject under discussion is the design of agricultural areas in cities, seen as aligned with the sustainable development of future cities. Urban agriculture and horticulture appeared in cities towards the end of the nineteenth century, but are currently regaining popularity, especially in Western European cities. This subject was discussed by Sroka and Musiał on a selection of cases [36]. They noted the significant social role of such solutions, their general acceptability, and a scale adapted to urban and suburban zones. New solutions in urban agriculture and horticulture are diverse and provide suitable conditions for experimentation and selecting optimal forms of development that consider local determinants and individual needs. They thus extend the recreational function of greenery to agriculture and this results in measurable utilitarian benefits.

## 3. Scope, Purpose, and Method of Research

The review of the literature indicates that issues concerning the impact of green areas, including pocket parks, on the quality of urban space are of an interdisciplinary nature. They thus form a crucial field for study, especially in relation to areas with high development and functiospatial structural density. Available publications on pocket parks feature findings that focus more on a local perspective of this subject—the form and function of the parks, their direct accessibility, and significance to the immediate surroundings. There is a lack of studies that would discuss the matter in a broader manner, e.g., by characterizing the impact of the concept of pocket parks and its implementation on an entire city or its larger fragment, especially relations between park layouts with the main public greenery and service system. This paper can contribute to filling in this gap in the state of the art. The subject of the impact of the construction of pocket parks on the quality of urban space appears to be important and topical, especially in a period of the densification of urban development, the extension of transport systems, and the increasing presence of nuisances in urban life. This study can be seen as a considerable contribution to the sphere of spatial and urban planning in terms of the improvement of the quality of life in the city, both on the local scale (that of a housing estate) and the supralocal scale, i.e., in relation to a district, or even a larger fragment of a city. The main objective of the study presented in this paper was to find an answer to the following questions:

- To what extent do built pocket parks affect the quality of urbanized space, via the local reinforcement of its utilitarian, aesthetic and environmental value?
- How does the idea of pocket parks contribute to improving the quality of the wider urban structure by forming supralocal linkages with the existing system of urban greenery and the system of public services?

- To what degree does the siting of a pocket park determine its local accessibility to users from the immediate vicinity, and to what extent do the layout of parks as built cover the demand for this form of space within a larger fragment of the urban structure?

This paper presents the findings of a study on the impact of pocket parks on the quality of urban space on the case of Krakow, Poland. Krakow is Poland's second largest city, with a population exceeding 800 thousand. It is a major academic, tourist, and economic center. The northeastern fragment of the city, which was selected for analysis, includes a part of the Nowa Huta district, as well as neighbouring districts dominated by multi-family housing. Nowa Huta, originally built after the Second World War, had originally been constructed as a separate city, and was only later administratively and structurally incorporated into Krakow. Nowa Huta's space was built following Socialist Realist ideas, but was planned in the spirit of pre-war Modernism, presenting a clear compositional layout along with a strongly developed public greenery system. At present, a green area development strategy is being implemented across all of Krakow, and features the so-called 'civic budget', which has led to the construction of numerous pocket parks within the city's urban structure. The subject of the study consisted of nine parks identified for study (eight existing and actively used parks and one that is under construction) located in the northeastern part of Krakowthe city, in the districts of Nowa Huta, Czyżyny, Mistrzejowice, Bieńczyce, and Wzgórza Krzesławickie. The names of these parks are as follows: Relaksacyjny (1), Teatralny (2), Lipowy (3), Prehistoryczny (4), Sąsiedzki (5), Polny (6), Sielski (7), Wiewiórkowy (8), and Niebiański (9). All of the parks are elements of a comprehensive programme of pocket park construction that is being implemented in Krakow—'Gardens for Cracovians'—by the municipality (Figure 1). The parks selected for investigation are located in the northeastern part of Krakow, within the city's administrative limits, within an area with a length of ca. 4.5 km (along the north–south direction) by 3.5 km (along the east–west direction), and all are public parks that were either built or were under construction in said area. The delimitation of the research site and the selection of mutually neighboring pocket parks was deliberate, as it allowed for a reliable analysis of the accessibility of the parks as a layout that was created and that functions within a greater urban structure. It also allowed for the investigation of the relations of a larger group of pocket parks with the system of public spaces and green areas of supralocal significance. Furthermore, the sites selected for analysis formed a representative sample, both in terms of size (9 out of 29 of Krakow's pocket parks) and diversity of features and determinants investigated.

According to data procured from the Municipal Greenery Authority in Krakow—a local government entity tasked with implementing the idea of pocket parks in Krakow—the 'Gardens for Cracovians' were by principle designed within this unit. Only a few parks were built based on designs prepared by 'external' designers and incorporated into the pocket park network upon completion. In every case, local communities participated in the design process. The form, use programme, and general development of a pocket park were determined by means of public consultations during the first phase of the programme, first for the entire system (2018) and then in relation to individual parks (as per their design and construction dates).

The study was conducted from two perspectives: a local (housing estate-level) one and supralocal one, referenced to a greater fragment of the urban structure (district, city fragment). The proposed perspective on the research problem points to the broad context of the pocket park idea—its local significance in urban space and its significant role in the shaping of the entirety of the urban structure and the interconnections between its elements. Studies of the functiospatial structure of cities point to different parts that co-determine the quality of this structure. These include: functional (a city's attractiveness and diversity in terms of uses, access to services), spatial (spatial structure integration, a city's compactness, hierarchical form), aesthetic (the cityscape's visual attractiveness, compositional and visual assets, public space aesthetics), technical (infrastructure, technical condition of development, and public spaces), environmental (access to green areas and water, a clean environment), social (intensity of interpersonal contacts, security, a well-

developed network of social spaces), and circulatory factors (mass transport efficiency, transport accessibility of key elements of the city's structure, ease of travel, walkability, and bikeability), among others. When engaging in the analysis and assessment of the impact of Krakow's pocket parks on the quality of urban space, the authors selected qualitative criteria that they saw as the most important from the perspective of the study on the local and supralocal scale. These are as follows.

- From the local perspective:
  - A pocket park's form and function as an indicator of a park's overall utilitarian and aesthetic attractiveness, which is crucial from the standpoint of a local user;
  - The development of a pocket park, created under the assumption of a new element of urban public greenery with consideration of the diversity of greenery forms introduced and a suitably large share of biologically active surfaces within the entire park;
  - The actual accessibility of each pocket park defined by the pedestrian access isochrone of about 10 min, and especially ensuring good accessibility to the widest possible group of users.
- From the supralocal perspective:
  - A pocket park's incorporation into the system of urban greenery and urban open areas of supralocal significance, i.e., determining the role of a park as an integral part of a larger system of greater natural significance to the city (integration of pocket parks with the main system of municipal public greenery);
  - Linking the park's location with the general system of urban public spaces and streets that act as service space sequences (integration of pocket parks with the system of main public spaces and services);
  - Accessibility of the entire system of pocket parks against the background of a larger fragment of the city, especially the investigation of the course of the 10 min pedestrian access isochrone of this system in reference to the location of major complexes of multi-family housing.

The quality criteria set by the authors for pocket parks can also be applied to other city structure elements, especially to public spaces such as streets and squares. It should be noted that the 'quality' in question is itself hard to define, as it depends on a range of different factors, some of which are strongly tied with subjective impressions and perceptions of space by individual users. It must be stated that the assessment of the attractiveness of individual pocket parks in terms of use, aesthetics, and from (from a local perspective), as presented, is subjective. It is also based on the results of opinion polls conducted by the Municipal Greenery Authority during the design stage of each park and its later construction and use in the form of public consultations (the formulation of expectations and preferences by residents as postulates for design briefs and the polling of user opinions about completed parks). The remaining qualitative criteria analyzed by the authors (associated with the individual accessibility of each park and the share of greenery in their development, as well as the general accessibility of the pocket park system and its links to the system of greenery and services in the city's wider structure) were conducted using graphical analysis methods, using available maps, satellite images, design documentation and digital tools. In Krakow, the study was based on a cycle of original field investigations of the parks and their surroundings (preparing a survey and photographic documentation). Materials on pocket parks in Krakow and on the 'Gardens for Cracovians' programme [37] were analyzed in combination with Krakow's planning documents, especially the Spatial development conditions and directions study prepared for the city [38]. A review of the literature was also performed. GIS tools and cartographic materials drawn to various scales were used in the study, including topographic maps and orthophotomaps of Krakow.

Over the course of the study, the authors decided not to administer questionnaires to the users of the parks. The assertation of the success of the idea of pocket parks in Krakow

was made based on the results of surveys conducted by the Municipal Greenery Authority, a public body responsible for the implementation of the entire concept in Krakow. The positive opinion of Krakow's residents of pocket parks is also evidenced by the fact that some of the parks were built as a result of so-called 'civic budgets' and can therefore be considered a grassroots initiative, inspired by local communities.

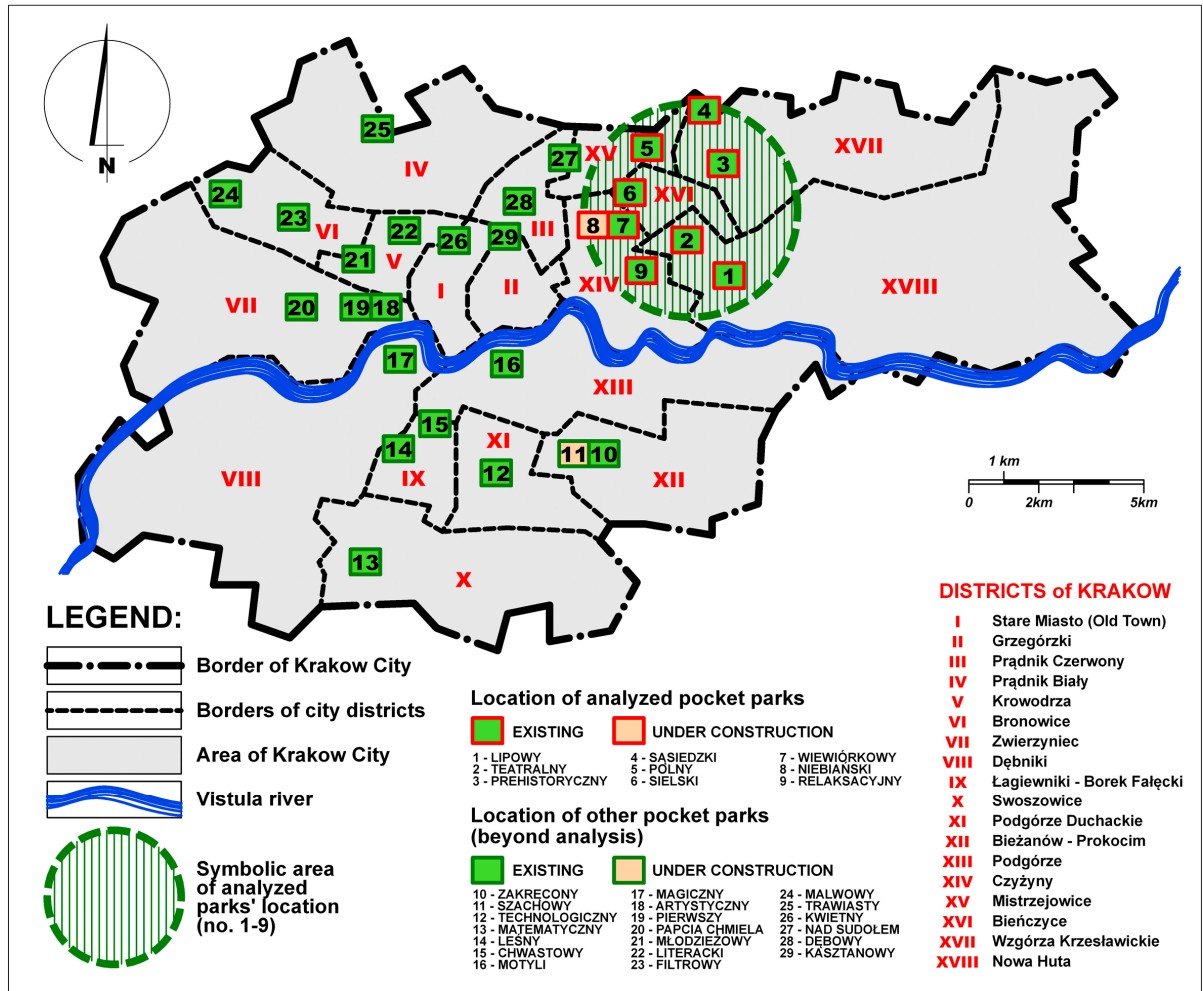

**Figure 1.** Location of pocket parks in Krakow set against the city's administrative limits and those of its districts (original work based on materials provided by the Municipal Greenery Authority in Krakow, displaying the state of completion of the 'Gardens for Cracovians' programme towards the end of 2022).

## 4. Results

The Krakow pocket parks selected for study were constructed in 2018–2022, while one is still under construction. It is thus a relatively new idea that has functioned in the area for a couple of years at most. It also appears that previously built pocket parks have gained public acceptance and are widely considered an element of green public space that is attractive in terms of form and function and that attracts potential users from the immediate surroundings. This is shown by the fact that during the period of original field research, the parks were frequented by large numbers of users who actively engaged with the available furnishing and development elements in these spaces. The group of parks under investigation is internally diverse. Overall, all the parks can be considered to be municipal public greenery, but each had an individual character (which was also expressed in the park's name). Apart from their overarching function being rest and recreation, the parks were used for educational, cultural, and sports-related purposes, depending

on their specific features and urban detail. The diversity of each park's characteristics, including its size, shape, and development, stems primarily from site-specific conditions, especially the form and function of the immediate surroundings. The study of the impact of pocket parks on the quality of urban space from a local perspective first included a detailed analysis of park types in terms of their size (area), geometric parameters (plan shape and dimensions), form of development, and scope of furnishing. Attention was focused on greenery solutions, i.e., the main form and structure of greenery introduced into the park's interior was determined, and each park's biologically active surface ration was calculated. Furthermore, the location of the parks was listed, as were their current state (either as currently in use or under construction) and year of opening. Each of the cases was illustrated with a representative photograph and a schematic floor plan showing the main dimensions. The analysis also covered the immediate and distant surroundings of the pocket parks in terms of forms of use and development structure located within 500 m from the site of each park. The results have been presented in Table 1 and shown in Figure 2.

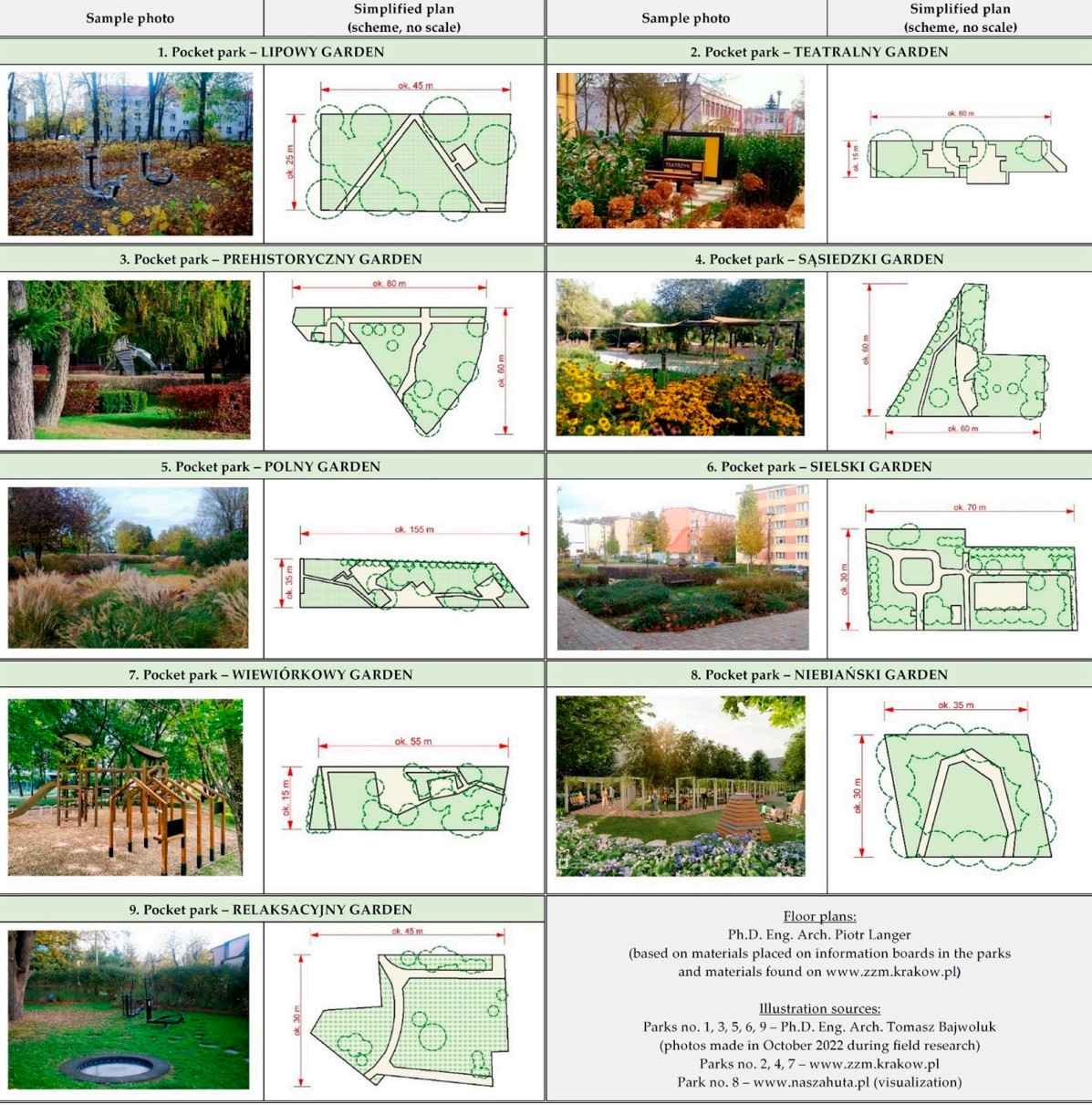

**Figure 2.** Comprehensive analysis of the selected Krakow pocket parks—photo and plan (original work).

**Table 1.** General overview of selected pocket parks in Krakow (original work).

| Item no. | 1. | 2. | 3. | 4. | 5. | 6. | 7. | 8. | 9. |
|---|---|---|---|---|---|---|---|---|---|
| Park Name | LIPOWY | TEATRALNY | PREHISTORYCZNY | SSIEDZKI | POLNY | SIELSKI | WIEWIÓRKOWY | NIEBIAŃSKI | RELAKSACYJNY |
| Location | Nowa Huta, J. Zachemskiego Street | Nowa Huta, Teatralne Housing Estate | Wzgórza Krzesławickie, Poległych w Krzesławicach Street | Wzgórza Krzesławickie, G. Morcinka Street | Mistrzejowice, Popielidów Street | Bieńczyce, W. Króla Street | Czyżyny, 2 Pułku Lotniczego Housing Estate | Czyżyny, Dywizjonu 303 Housing Estate | Czyżyny, F. Weżyka Street |
| Current state and opening year | Existing, open 2018 | Existing, open 2020 | Existing, open 2020 | Existing, open 2020 | Existing, open 2020 | Existing, open 2019 | Existing, open 2022 | Under construction * | Existing, open 2019 |
| Plan dimensions m × m (approximate) | 45 × 25 | 60 × 15 | 60 × 80 (max.) | 60 × 60 (max.) | 35 × 155 | 30 × 70 | 55 × 15 | 35 × 30 ** | 25 × 25 (max.) |
| Plan shape | regular, rectangular | regular, rectangular | irregular, polygonal | irregular, polygonal | regular, trapezoid | regular, rectangular | regular, rhomboid | regular, rhomboid | irregular, polygonal |
| Area, ha/m² | 0.11/1100 | 0.09/900 | 0.32/3200 | 0.17/1700 | 0.46/4600 | 0.21/2100 | 0.10/1000 | 0.09/900 | 0.13/1300 |
| Development and features | paths/alleys, benches, swings, educational elements, sports equipment | paths/alleys, table, benches, audience stand, theatre, play features | paths/alleys, platforms/terraces, pergolas, seating/benches, play features | paths/alleys, platforms/terraces, pergolas, low fences, seating, hammocks, flower containers | paths/alleys, educational elements, benches, sports equipment | paths/alleys, benches, pergolas, sports pitch, play features, low fences, sculptures | paths/alleys, play features, benches, tables, educational elements, sculptures | paths/alleys, play features, platform, benches, table, hammock, pergola ** | paths/alleys, benches, recliners, swing, sports equipment |
| Overall form and development of park greenery | singular trees, tree groups, hedges, flowerbeds, lawns | tree groups, bushes, lawns | singular trees bushes flowerbeds lawns | tree groups, singular trees, flowerbeds, low plants in containers, lawns | singular trees tree groups bushes, vines flowerbeds lawns | singular trees tree rows flowerbeds | tree groups, flowerbeds lawns | tree groups tree screen flowerbeds lawns ** | singular trees lawns flowerbeds |
| Ratio of biologically active surface to total park area | ca. 85% | ca. 60% | ca. 80% | ca. 85% | ca. 70% | ca. 75% | ca. 70% | ca. 85% ** | ca. 80% |
| Overview of immediate surroundings | multi-family housing (detached), car park | theatre, multi-family housing | multi-family housing, city park, service building | playground, multi-family residential building, wasteland car park | multi-family housing, production building | multi-family residential building, commercial and service buildings, car parks | multi-family residential building, car park, housing estate greenery | church, car park multi-family residential building | multi- and single-family housing, service building |
| Overview of distant surroundings *** | multi-family housing estates, commercial services and other public buildings, hospital, partially landscaped greenery | multi-family housing estates, commercial services and other public buildings, partially landscaped greenery | multi-family housing estates, service buildings, city park, cemetery, allotment gardens | Single-family housing estates, allotment gardens, agricultural areas, post-industrial areas, wasteland | multi- and single-family housing estates, allotment gardens, agricultural areas and wasteland, circulation spaces | multi-family housing estates, commercial and service buildings and complexes, circulation spaces | multi-family housing estates, commercial services, offices, and public buildings, landscaped areas | multi-family housing estates, commercial services, offices, and public buildings, landscaped greenery | multi- and single-family housing estates, service buildings and complexes, circulation spaces, industrial plants |

* Estimated opening date 2023; ** based on design documentation; *** within an area demarcated by a 10-minute pedestrian accessibility zone (500 m).

Over the course of research conducted from a local perspective, a graphical analysis of the pedestrian accessibility of each pocket park was performed. The boundary value for a park's 'good' accessibility was an isochrone of 10 min pedestrian access, which is about 500 m—assuming a slow movement pace (e.g., that of a senior or small child)—which is compliant with the assumptions of the 'Gardens for Cracovians' programme. The good accessibility range for each park was geometrically determined and presented on a map, accounting for the actual travel time using existing pedestrian paths and the course of circulation barriers that limit accessibility. The graphical interpretation of the analysis of the pocket parks' pedestrian accessibility has been presented in the form of schemes in Figure 3.

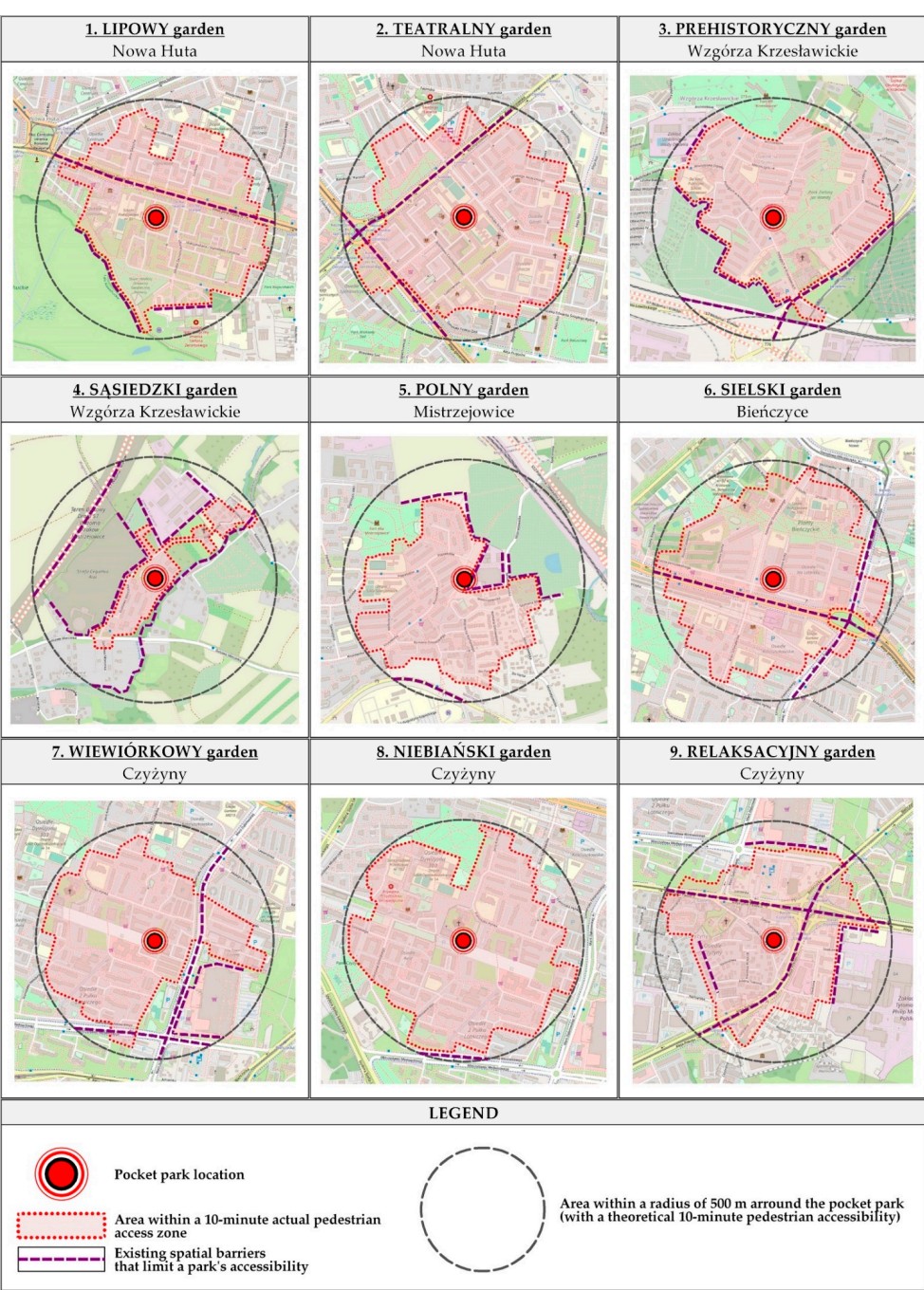

**Figure 3.** Pocket park accessibility analysis within a 10 min pedestrian access isochrone—schemes (original work).

The study of the impact of pocket parks on the quality of urban space from a supralocal perspective focused on the geometry of the entire layout under analysis, defined by the park sites. The analysis enabled the determination of interrelationships between neighboring pocket parks and the spatial relations between the investigated park complex and the overall system of public greenery and the system of major public spaces and services, including streets that act as sequences of commercial spaces (defined in the current Spatial development conditions and directions study for Krakow, which the authors saw as reflecting the current and future shape of this system). The analysis performed from the supralocal perspective considered the placement and fundamental form of green areas and major services, as well as the location of large complexes of multi-family housing, large industrial areas, and the citywide transport layout, including planned arterials (Figure 4).

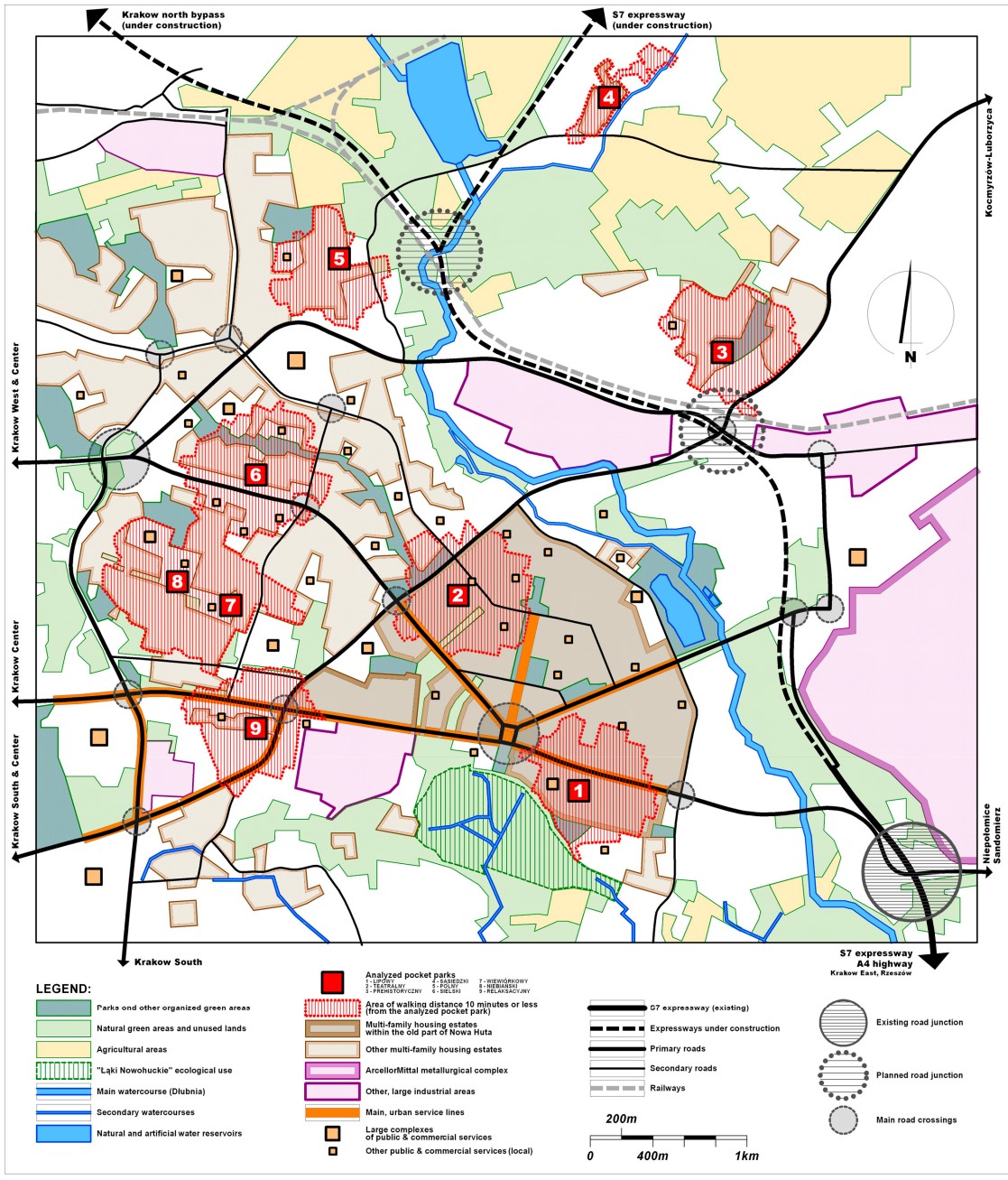

**Figure 4.** Analysis of selected pocket parks from a supralocal perspective—relations with the system of greenery and accessibility of major public spaces (original work).

In terms of determining the accessibility of the layout of pocket parks under study within a larger area of the city, the accessibility ranges of all the parks were shown summarily, which allowed the tracing of the areas of multi-family housing with good access to existing public parks, and those which have limited access, namely, with a travel time greater than 10 minutes' walking distance.

The analyses showed that the pocket parks are 'parks' and 'gardens' not only in name, but also in terms of development and the formal diversity of their greenery designs. Despite occupying a relatively small area (from 0.09 ha to 0.46 ha), biologically active surfaces amount to between 70 and ca. 85% of the parks' surface (it was 60% in only one case). The wide range of greenery design solutions is also notable, as it includes both singular and grouped trees (in both loose and structured layouts), shrubs, hedges, and vines, as well as diverse, multicolored flower formations, tall grasses, and mowed lawns.

The local-level analysis also found significant differences in the accessibility of each park. In general, almost all the parks were located in high-density areas predominated by multi-family housing. It can thus be assumed that the sites of the parks provide relatively good access to greenery to a potentially large group of users (predominantly residents). This condition was met to the greatest extent in the cases of Teatralny Park (2), Sielski Park (6), and Niebiański Park (8). In a couple of cases, park accessibility was severely limited by the presence of difficult-to-overcome transport barriers: busy arterials or extensive gated areas (industrial plants, allotment gardens, cemeteries, etc.). The analysis of the extreme example of Sąsiedzki Park within the Wzgórza Krzesławickie housing estate (4) showed that the pocket park had a severely limited accessibility, and its potential users were almost exclusively the residents of the small Zesławice housing estate. Similarly, Rekreacyjny Park in Czyżyny (9) had its 10 min pedestrian access zone covering mostly an area of detached single-family buildings (houses with yards), i.e., areas whose residents are unlikely to use the pocket park, despite having good access to it.

The supralocal-level analysis allowed for investigating existing spatial relations between the sites of the pocket parks (understood as a layout of 9 isolated elements) and the overall system of municipal public greenery and the main system of public spaces and services whose significance extends beyond their local areas. These relations were identified independently for each park, defining its integration ('integrated' or 'unintegrated') separately relative to the greenery system and service system. The results of the analysis have been collectively presented in Table 2.

The analysis performer in this paper clearly showed that the integration level of the pocket parks investigated with major systems of municipal greenery and public services was varied. The parks located in Nowa Huta—Lipowy (1) and Teatralny (2)—can be considered well-integrated with the municipal system of green and service areas, which is highly significant from the standpoint of urban structure quality when discussed from a supralocal perspective. Incorporating a pocket park into the wider layout of public spaces enhances the diversity and thus the utilitarian and aesthetic attractiveness of the entire system. It is also beneficial in terms of integrating the city and the functioning of the natural system inside the urban structure, especially when it is densely developed. On the other hand, Prehistoryczny (3), Sąsiedzki (4), Wiewiórkowy (7), and Niebiański (8) parks are separated from the major urban greenery and service system due to site-specific conditions and the proximity of transport barriers, primarily two-carriageway arterials and gated industrial sites. These parks have the forms of clearly demarcated enclaves of essential local significance but have a marginal supralocal role. It should also be stressed that in the area confined to the 10 min pedestrian access zone of all the parks, there were either singular service or public greenery areas; however, these elements were not a part of the overall urban system, being of only local significance.

In terms of the accessibility of pocket parks, discussed from a supralocal perspective, the results of the analysis lead to the conclusion that the layout of parks under investigation does not guarantee access to potential users from many multi-family housing complexes. The graphical interpretation of this accessibility shows that a significant part of the housing

estates of Nowa Huta, the eastern zone of Bieńczyce and Czyżyny, as well as a large of Mistrzejowice remained outside of the 10 min pedestrian access zone of pocket parks that have thus far been built in Krakow's urban structure (Figure 2). On the other hand, three of the analyzed parks—Sielski (6), Wiewiórkowy (7), and Niebiański (8)—were found to be located quite close to each other, and the areas covered by their pedestrian access isochrones partially overlapped. This means that the area's residents have very good access to more than one pocket park, while some parks of the group under study have strong functional linkages.

**Table 2.** Level of integration of the investigated pocket parks with the municipal greenery and services system (original work).

| Item No. Park Name | Degree of a Pocket Park's Integration with the Main System of Public Greenery | Degree of Integration of a Pocket Park with the Man Public SPACE System |
|---|---|---|
| **1. LIPOWY garden** | **INTEGRATED** located close to the public greenery system * | **INTEGRATED** located close to urban commercial space sequences and major services * |
| **2. TEATRALNY garden** | **INTEGRATED** located close to the public greenery system * | **INTEGRATED** located close to urban commercial space sequences and major services * |
| **3. PREHISTORYCZNYgarden** | **NOT INTEGRATED** located at a significant distance from the public greenery system ** | **NOT INTEGRATED** located at a significant distance from urban commercial space sequences and major services ** |
| **4. SSIEDZKI garden** | **NOT INTEGRATED** located at a significant distance from the public greenery system ** | **NOT INTEGRATED** located at a significant distance from urban commercial space sequences and major services ** |
| **5. POLNY garden** | **INTEGRATED** located close to the public greenery system * | **NOT INTEGRATED** located at a significant distance from urban commercial space sequences and major services ** |
| **6. SIELSKI garden** | **INTEGRATED** located close to the public greenery system * | **NOT INTEGRATED** located at a significant distance from urban commercial space sequences and major services ** |
| **7. WIEWIÓRKOWY garden** | **NOT INTEGRATED** located at a significant distance from the public greenery system ** | **NOT INTEGRATED** located at a significant distance from urban commercial space sequences and major services ** |
| **8. NIEBIAŃSKI garden** | **NOT INTEGRATED** located at a significant distance from the public greenery system ** | **NOT INTEGRATED** located at a significant distance from urban commercial space sequences and major services ** |
| **9. RELAKSACYJNY garden** | **NOT INTEGRATED** located at a significant distance from the public greenery system ** | **INTEGRATED** located close to urban commercial space sequences and major services * |

* At a distance that provides pedestrian access within 10 min; ** at a distance that exceeds pedestrian access within 10 min.

## 5. Summary

Creating the best possible conditions for human life and functioning in cities is a major challenge. Research on urban structures shows that ensuring direct contact with nature, even on a small scale, is a key aspect of the sustainable and rational planning of space [8]. One practical action in this regard is the construction of pocket parks which are an interesting form of providing and organizing green areas in densely developed areas by using small, cameral spaces. It should thus be assumed that the introduction of

pocket parks into the interiors of urban structures can exert a specific impact on the state of an entire city or its major fragment [10,11]. In the definition of the research problem, the authors defined criteria that are essential to the quality of urban space, discussing them from two perspectives: the local and the supralocal. This allows for the analysis and assessment of the pocket parks themselves as isolated, cameral spaces, and enables the investigation of relations between individual parks and determining their role in the overall system of urban greenery. The presented perspective on the problem is largely compliant with the method and scope of research conducted by others, including studies of Beijing, China, ref. [2] or Kuala Lumpur, Malaysia [25]. These studies covered a total of nine pocket parks that have been recently built in an area of Krakow. Over the course of the study, a detailed analysis of each park was performed, and an investigation was conducted into their mutual relations and the link between the existing layout of pocket parks as well as other key elements and the urban structure on the supralocal scale, in the context of the public greenery system and the layout of major public and service spaces.

It should be noted that this study and paper concerns only a portion of Krakow's pocket parks constructed as a part of the comprehensive 'Gardens for Cracovians' programme. It appears justified to perform a holistic analysis and assessment of the impact of the project's completion on the quality of urban space both citywide and as a reference for other cities, including those outside Poland and of a similar size to Krakow.

The authors wish to stress that issues associated with shaping urban greenery systems in urban structures are not equivalent to urban ecology. This is supported by numerous studies, including those by Picket et al. (2016) [39], Gaston (2010) [40], or Rebele (1994) [41], as well as many others. It is also difficult to deny that green areas, including pocket parks, are a part of a city's environmental system despite having anthropogenic origins, and that their environmental value is undeniably lower than that of primal, untransformed elements of nature. The authors intended this paper to engage with issues of urban space quality in reference to the pocket park system—their form, function, and local accessibility (as factors that define their quality on the local scale)—and its general accessibility and linkages with the citywide system of greenery and services (as factors that determine quality on the supralocal scale). It was not the objective of this paper to present the problem from an 'ecological' standpoint, but from that of architecture and urban planning, in line with the academic discipline represented by the authors. Therefore, the research results presented are aligned with the interdisciplinary character of issues associated with pocket parks. The research current represented by the authors does not run counter to 'eco-oriented' perspectives on the subject; both perspectives can interweave and positively complement each other.

## 6. Conclusions

This study of Krakow's pocket parks in terms of their impact on the quality of urban space on the local and supralocal scale allowed for the formulation of the following conclusions:

- Analysis of the form and development of the pocket parks showed their high utilitarian and aesthetic attractiveness, as well as their positive impact on the quality of housing space (on the local scale). The observed intensity of the parks' use allows us to assume that the initiative to build them has gained social acceptance and should be developed in the future;
- The key factor in implementing the idea of 'pocket parks' is not their number, but their siting, layout, and mutual linkages. From a local perspective, the criterion of siting determines the number of potential users who are located within the zone of comfortable access to the parks; this can be investigated in every case, accounting for the space's development and the presence of circulation barriers around a park's site. From a supralocal perspective, the locations of each pocket park form a system that generates specific mutual relations between the parks and determines the linking (integration) of the parks with the system of urban greenery and the main system of services-based public spaces;

- In the light of the above, it can be beneficial to consider local determinants that define a park's accessibility each time when siting a park, while also investigating spatial relations on a wider scale, especially in terms of linkages with the greenery and services system, which creates the opportunity to incorporate pocket parks into the network of spaces that integrate and permeate larger areas of the city, enhances their accessibility, and increases their diversity and overall attractiveness;
- The research method presented in this paper can be useful in analyzing existing, already constructed pocket parks, and can also be applied in the future development of this concept, both in Krakow and in other cities. The authors believe that making the decision on the siting of a pocket park in each case requires a prior analysis of said park's accessibility and an identification of its spatial relations with the overall public greenery and services system, which appears crucial from the standpoint of the quality of urban space.

**Author Contributions:** Conceptualization, P.L. and T.B.; Methodology, P.L. and T.B.; Software, P.L.; Validation, P.L. and T.B.; Formal analysis, P.L. and T.B.; Investigation, P.L. and T.B.; Resources, P.L. and T.B.; Data curation, P.L. and T.B.; Writing—original draft, P.L. and T.B.; Writing—review & editing, P.L.; Visualization, P.L.; Supervision, T.B.; Project administration, T.B.; Funding acquisition, P.L. and T.B. All authors have read and agreed to the published version of the manuscript.

**Funding:** This research received no external funding.

**Institutional Review Board Statement:** Not applicable.

**Informed Consent Statement:** Not applicable.

**Data Availability Statement:** Not applicable.

**Conflicts of Interest:** The authors declare no conflict of interest.

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
