# Peer review of "The Pocket Park and Its Impact on the Quality of Urban Space on the Local and Supralocal Scale—Case Study of Krakow, Poland"

_sustainability, doi:10.3390/su15065153_

Round 1

Reviewer 1 Report

The manuscript deals with a relevant content, is clearly structured and written in an understandable way. Before it is suitable for publication, the following changes should be made:

1) In order to highlight the topicality of the topic, the relevance of the topic with regard to major societal challenges should be formulated more explicitly (such as climate change, societal individualization, re-urbanization).
2) Krakow is certainly suitable as an example city. However, its suitability should also be made explicit.
3) Krakow is a very important city for Central Europe and Poland in particular. However, the sustainability is aimed at an audience that is also located outside Europe. In this respect, general information on the location, size, history (for example, as a post-socialist city, with the specific feature Nowa Huta) would be very desirable for the classification.
3) The authors clarify their understanding of quality, which is often omitted. This is very useful. In order to make the contribution more connectable, it would be desirable to emphasize this even more, for example by comparing one's own approach with alternative understandings. Also, the own approach could be placed more strongly in the tradition of a user-oriented evaluation.
4) In the conclusion, it could be made clearer which findings the authors consider central and new in the context of the current state of research (e.g., in relation to points 1 and 3 above).

With the exception of these possibilities for improvement, which are conducive to the dissemination of the article, I consider the article to be very successful.

Author Response

Thank you very much for your comments in the received review. We have included them in the article by changing and supplementing its content.

  1. This is a very important remark regarding the essence, timeliness and importance of green areas, especially in the field of counteracting climate change. The note has been included in the text.
  2. The authors referred to the choice of the study area - Krakow as a place for the implementation of pocket parks, in accordance with the strategy of development of green areas of the city.
  3. The necessary information for Krakow as the area of research was introduced.
  4. The concept of space quality in other city structures was expanded and clarified.
  5. The summary clarifies the meaning of the adopted method.

Reviewer 2 Report

The research theme is interesting and the paper is quite well-written. However, I have some suggestions in order to improve the paper:

 ·         I suggest shortening the abstract, as it needs to be concise and straight to the point.

·         In the introduction, there are some statements that need a proper referencing:

Example:

The idea to create cameral green enclaves called pocket parks originated in North America and Europe towards the end of the 1960s. It stemmed from contemporaneous attempts at finding a new form of public space that would be widely accepted and oriented towards satisfying the needs of housing communities in terms of using organized, local green areas for rest and recreation. The idea of the pocket park was also aligned with the notion of sustainable urban development by providing easy access to natural areas, including landscaped and natural greenery.

I suggest having separate chapters – Introduction (where the authors will introduce the topic and explain the research aims and the need for research, as well as contributions) and Theoretical framework (Literature review where the authors will analyze the current knowledge on this topic). Having both in the introduction is not comprehensive and clear enough.

The methods and materials need more info about the method used (in several sentences)

I suggest that authors add a few sentences about how this topic is connected with sustainability, preferably in the introduction

The paper lacks a discussion of the findings and connection with the existing literature.

Finally, please add study limitations and directions for future research. Emphasize the study contributions explicitly.  

Author Response

Thank you very much for your comments in the received review. We have included them in the article by changing and supplementing its content.

  1. The abstract was corrected.
  2. The authors refer to the history of the idea of pocket parks in the world in the context of its application in Polish cities only in recent years. This indicates the relevance of the idea and its timeliness.
  3. The authors carefully considered the suggestion of separating the introduction and review of the literature on the subject - a decision was made to leave the current arrangement.
  4. The authors referred in the text to the relationship of the discussed topic with the sustainable development of cities. They emphasized that the applied research method shows a broad context of using the idea of a pocket park in the city space. In the summary, the links with the literature were supplemented and the original contribution to the research on the topic was emphasized.

Reviewer 3 Report

Why inserting Table 1 and Table as pictures.

Be consistent in the use of text and words “Fig 2” and “Figure 2”throughout the manuscript. Stick to one, preferably, “Figure 2”

The title of a table (Table caption) should appear above the Table and not below the Table. Please attend to this!

Avoid use of different font types and sizes in Tables and Figures. Please correct this throughout the manuscript.

A conclusion should not appear as if it’s a summary. A conclusion should highlight the implication or implications of the research findings and the research methodology developed in this study.

Author Response

Thank you very much for your comments in the received review. We have included them in the article by changing and supplementing its content.

  1. Captions and table markings have been corrected. The style and typeface of the tables and text have been unified, with the proviso that the remaining variation in font size results from the formatting of the tables and their sizes.
  2. The remark was taken into account - the summary and conclusions from the research were separated.

Reviewer 4 Report

The research deals with a topic of definite ecological-environmental importance, namely, urban green areas. Nevertheless, the study is oriented toward a treatment of the topic from an architectural point of view, in line with the authors' background. In my opinion, the research misses the focal point of the journal by not dealing at all with the ecological aspects of urban green areas, the ecosystem services they provide, of which improving well-being and quality of life are an integrated part, green corridors, and ecological networks, heat island mitigation, etc. Submission of work in a landscape architecture journal is recommended.

Author Response

Thank you very much for your comments in the received review.

The research issues raised in the article refer to the broadly understood sustainable development of cities. It focuses on the basic element of the natural environment in the city, which is green areas, including pocket parks. According to the authors, the presented issues, the scope of research and their results closely refer to the "ecological" approach to the problem of shaping the spatial and functional structure of the city, especially since they take into account the importance of pocket parks in a comprehensive system of green spaces. For this reason, this article is part of the journal's profile.

Round 2

Reviewer 1 Report

Dear authors, the article should now get the resonance the topic deserves due to the changes. Best regards

Author Response

The authors would like to thank the Reviewers for the time and effort spent on reviewing the paper and confirm that the paper has been revised according to all of the Reviewers’ comments. In addition, as a result of the second phase of the review, the paper’s content has been revised again.

Reviewer 2 Report

Dear author, 

It is very disappointing to see that you did not consider the majority of the comments and suggestions I provided in order to improve your paper. 

Therefore, I suggest that the paper is not accepted till consider the necessary changes to bring the paper to the publishable format. 

Author Response

The Reviewer’s comments are largely similar to the suggestions and reservations of the Academic Editor and have been fully included in the second phase of the paper’s revision.

  • The abstract has been shortened and the paper’s title has been revised to better tie it in with the research field.
  • As per the Reviewer’s suggestions, the introduction and state of the art section have been separated and the link with sustainability has been explained in the state of the art section.
  • Information on the method has been included in the introduction.
  • The conclusion section has been revised to include references and the direction of further research, contributions have also been added, assuming criteria that are essential to the quality of urban space and have been considered from two perspectives: local and supralocal.

Reviewer 3 Report

Much improvement of the paper has been noted

Author Response

(The authors gave the same response as above.)

Reviewer 4 Report

The authors are reminded, first of all, that urban greenery cannot refer to the natural environment of the city, because there is no natural environment, properly understood, in the city, and secondly, urban greenery is the result of human labor, thus something far removed from any concept of naturalness.

Moreover, the treatment of the "problem of the formation of the spatial and functional structure of the city," even when the results "take into account the importance of pocket parks in a complete system of green spaces," DOES NOT IN ANY WAY CONSTITUTE AN ECOLOGICAL APPROACH of the issue, because the ECOLOGY OF GREEN AREAS IS A VERY DIFFERENT THING, and there is no trace of it in the research presented.

A reading of ecology articles dealing with green areas might help to understand the remoteness of the research from ecological issues, for example, Elgizawy, 2014; Li et al., 2017; Bolund & Hunhammar, 1999; Burkhard et al., 2012....

IT IS NOT ENOUGH TO TALK ABOUT GREEN AREAS TO CLAIM TO BE TALKING ABOUT ECOLOGY.

Author Response

The authors fully agree with the Reviewer that issues of urban greenery system design in urban structures are not equivalent to ‘urban ecology’. This is proved by the references provided by the Reviewer, as well as findings presented in other studies (e.g., Picket et al., 2017 – Evolution and future of urban ecological science: ecology in, of and for the city, Gaston, 2010 – Urban ecology; Rebele, 1994 – Urban ecology and special features of urban ecosystems and many others). It is also difficult to deny that green areas, including pocket parks, belong to a city’s natural system despite having an anthropogenic origin and their environmental value is undeniably lower than that of fully natural, untransformed environments. The authors intended the paper to discuss issues of the quality of urban space as referenced to the system of pocket parks – their form, function and local accessibility (as factors that determine quality on the local scale), as well as general accessibility and linkages with the urban greenery and service systems (as factors that determine supralocal quality). It was not the paper’s objective to present this problem from an ‘ecological’ perspective, but from an ‘architectural and urban’  view, in accordance with the academic discipline represented by the Authors. In addition, the main objective of the publication the Authors have submitted their manuscript to is sharing experiences between researchers from different fields so as to engage in interdisciplinary cooperation. For this reason, it is highly justified to consider research on shaping urban greenery systems and their quality informed by the perspective of architects and urban planners. It is worth noting that the research presented in the paper does not conflict with an ‘environmental’ perspective on the subject, and both perspectives can interweave and complement each other.

Round 3

Reviewer 2 Report

Dear authors,

Thank you for revising the paper according to my suggestions. Please note that introduction part should end with study objectives, so they should be moved from methodology to Introduction. Scope of the paper is never introduced so late (in method section), but always in Introduction.

Reviewer 4 Report

The manuscript does not have a solid ecological basis